# Conflicts in Implementing Environmental Flows for Small-Scale Hydropower Projects and Their Potential Solutions—A Case from Fujian Province, China

**Qizhen Ruan** [1,2,3], **Feifei Wang** [1,2,3] **and Wenzhi Cao** [1,2,3,*]

1 State Key Laboratory of Marine Environmental Science, Xiamen University, Xiamen 361102, China; ruanqizhen@stu.xmu.edu.cn (Q.R.); feifeiw@xmu.edu.cn (F.W.)

2 Key Laboratory of the Ministry of Education for Coastal Wetland Ecosystems, Xiamen University, Xiamen 361102, China

3 College of the Environment & Ecology, Xiamen University, Xiamen 361102, China

* Correspondence: wzcao@xmu.edu.cn

**Abstract:** Releasing environmental flows is a valuable strategy for mitigating negative impacts of small-scale hydropower projects on river and riparian ecosystems. However, maintaining environmental flows has faced considerable resistance from different stakeholders, and previous studies have failed to appropriately investigate solutions. Here, online questionnaires and interviews were conducted among small-scale hydropower project owners, government administrators, and the public in Fujian Province, China. The results showed that the major hindrance to implementing environmental flows was the potential economic loss resulting from reductions in electricity production, stakeholders' skepticism, technical difficulties, and a lack of the government supervision. Diversion-type projects pose the largest losses of electricity production after the release of environmental flows, and by adopting a 10% of mean annual flow as minimum target, most small-scale hydropower projects obtain low marginal profits without compensation. Here, we proposed an appropriate payment for ecosystem services by introducing an economic compensation program for different types of small-scale hydropower projects scaled by potential losses in electricity generation. Under such a scheme, economic losses from a reduction in electricity production are covered by the government, hydropower project owners, and electricity consumers. Our study offers recommendations for policymakers, officials, and researchers for conflict mitigation when implementing environmental flows.

**Keywords:** conflicts; environmental flows; small-scale hydropower projects

## 1. Introduction

Hydropower is the most common renewable energy source for electricity production. Small-scale hydropower projects (SHPs) play an important role in generating electricity and have been established in 166 countries [1], of which China had ranked first with over 47,498 SHPs by the end of 2017. SHPs in China are defined as having an installed capacity under 50 MW, although there is no internationally agreed definition [2]. The Chinese government encourages the development of renewable energy, such as hydropower and wind power, from which all electricity is purchased by grid companies. There is no unified feed-in tariff for SHPs in China, and each province has the right to set its benchmark price, which is based on SHP development costs and the average purchasing price of the provincial electricity grid company [3].

In the last decade, more attention has been paid to the ecological impacts induced by SHPs, such as hydrological alteration [4–6], river connectivity fragmentation [7], habitat losses [8], and changes in species composition [9,10]. Research has also highlighted the

cumulative impacts of SHPs to gain a better understanding of their environmental consequences [11–13]. One important impact is the alteration in natural flow regimes, including river flow depletion [14], which has been proven to be related to the type of hydropower [15–17]. In this study, SHPs were grouped into three categories, namely diversion-type, barrier-type, and mixed-type projects. Both diversion-type and mixed-type projects transfer flow away from natural watercourses through channels or pipelines [18]. Barrier-type projects can be further classified into run-of-river projects and reservoir-type projects depending on their mode of storage. Diversion-type projects are most likely to dry up flows, especially during the dry season [19,20].

Environmental flows (E-flows) refer to discharge volumes that should remain in the river channel [21] to sustain freshwater ecosystem health and human well-being [22]. For the last 50 years, numerous studies have assessed E-flows for ecological health [23–28]. Not until this century have E-flows gradually been incorporated into legislation and regulation practices in many countries [29]. However, in many cases, E-flows are still at the stage of discussion and policy enactment [30], while their implementation faces political, economic, technical, and social challenges [31,32]. We stress that there is a disconnection between booming E-flows science and practice. Currently, there is insufficient work that integrates practice into E-flows literature. Of the existing narratives, there exists a lacuna in mitigating the conflicts in E-flows implementation for SHPs, especially those that incur losses, and the issue of "willingness to pay" [33].

In China, to operate SHPs, one needs an environmental impact assessment and electric power business license. However, E-flows were not involved in environmental impact assessment until the first official requirement of E-flows was stated in 2006 [33]. Additionally, due to neglect of the environmental impacts of SHPs, the regulation only proved effective for large-scale hydropower projects. As the first province in China to enforce E-flows implementation for SHPs, the Fujian provincial government made little progress in implementation until the Jiulong River experienced algal blooming in 2009. This problem was finally solved by opening the sluice gates of all the upper stream hydropower projects. In addition, there are more than 6000 licensed SHPs in Fujian Province [34], including diversion-type (76.7%), barrier-type (11.8%), and mixed-type (11.3%). Crucially, most SHPs in Fujian Province lack the necessary facilities for releasing E-flows because the majority (99.7%) of SHPs had been established before the first Chinese regulation of E-flows was issued.

Discharge and flow velocity are critical factors affecting algal blooming [35,36], which occurs more frequently in rivers with more hydropower projects in Fujian Province. To prevent algal blooming, the Provincial Department of Environmental Protection has required the SHPs of 12 primary rivers to release E-flows and install online monitoring facilities in 2009 [37]. Implementation has involved two different methods, with either "10% of Mean Annual Flow" (10%MAF) [38] or "90-percent exceedance probability of the average flow rate in the driest month based on statistics of monthly mean flows at least 10 years" (Qdm90) as the minimum target [38]. Limited by hydrological data, 10%MAF was used for SHPs in rivers with a drainage area of < 500 km$^2$, which account for 85% of the total SHPs [39], while Qdm90 was adopted by SHPs on the main channels with a drainage area > 500 km$^2$ [40]. However, at the end of 2010, only 28% of the 415 required SHPs had been installed with monitoring facilities [41].

In response to those limitations of the existing literature and urgent demand of releasing E-flows, here, we provide a case study in support of recommendations to facilitate the implementation of E-flows for SHPs. This study is the first known attempt to gather perspectives on E-flows, SHPs, and willingness to pay from three interest groups based on questionnaires and interviews. The objective of the study was to determine the key conflicts in implementing E-flows and to propose potential solutions. By reviewing the literature [42–44], three factors were selected as the main obstacles, namely economics, stakeholders' skepticism, and technologies. Here, we define economic conflicts as eco-

nomic losses induced by retro-fitting dams and releasing E-flows; stakeholders' skepticism encompasses differences in opinion on whether SHPs are green and the necessity of implementing E-flows; and technical difficulties that include the engineering feasibility of retrofit dams for releasing E-flows. We hypothesized that the threat of economic losses would contribute the most to these potential conflicts, as has been previously suggested in the literature [31,44,45]. We also examine stakeholder perspectives on who should pay for incurred losses and their willingness to pay. Specifically, we aimed to (1) explore the environmental impacts of SHPs in Fujian Province, (2) analyze the difficulties and stakeholder conflicts when implementing E-flows, and (3) examine the current mitigation measures and propose potential solutions.

## 2. Materials and Methods

### 2.1. Questionnaires

Online questionnaires were sent to a random sample from members of the public of over 18 years old in Fujian Province. Snowball sampling was adopted to distribute online questionnaires to SHP owners and related government administrators in Fujian Province with the help of the Fujian Province SHP Association by online links because it is recommended for use when samples are rare and difficult to find. There was no preference for selecting respondents in each group. Questionnaires were solely comprised of closed-ended questions with either response at the nominal level or binary response (see Table S1). The sample questions of single choice (A.) and multiple choices (B.) are shown as follows: (A.) Do you think it is necessary to implement environmental flows? (Yes/No); (B.) Who needs to bear the economic loss generated by implementing environmental flows? (The government solely/The owners of SHPs/Electricity consumers by paying more for the electric bill/The government and the owners/The government and electricity consumers/The government, the owners, and electricity consumers).

As it is impossible to know how many times the online questionnaire links had been clicked, we were only able to filter invalid questionnaires by setting up reverse questions; if the obverse and reverse choices were selected at the same time, the questionnaire was considered invalid. The purpose of the study was presented before the questions to ensure each respondent was informed. After a pretest, the question template was re-evaluated; some questions were explained, and some were simplified. The number of questions posed to each target group was different (nine for SHP owners, 11 for government administrators, and six for the general public). All of the questionnaires focused on the environmental impacts of SHPs, attitudes towards SHPs as green enterprises and the E-flows release, perspectives on payment for ecosystem services (PES) as a cost-sharing program, and the willingness to pay for E-flow implementation. The questions to government administrators and owners also covered the conflicts and difficulties of E-flows implementation, average returns and electricity production losses and views on existing compensation policy.

A total of 513 owners, 58 government administrators, and 667 members of public completed the questionnaires, with corresponding validity rates of 93% (478), 93% (55), and 90% (603), respectively. These high validity rates likely reflect the fact that all respondents volunteered to complete the questionnaires, i.e., people with a low willingness to respond would ignore the original links. The chi-square test was adopted to examine the differences in the choices of respondent groups, where $p < 0.05$ indicated a significant difference.

Yet, respondent accessibility has the potential to affect the response rate and prejudice the results, particularly in a survey targeted at a large area. Other survey limitations include gathering responses from those who did not actively participate.

*2.2. Interviews*

Semi-structured interviews were conducted after the questionnaires were collected to obtain a more comprehensive understanding of the perspectives of different interest groups. These semi-structured interviews were flexible and allowed interviewers to alter the pace and order of questions depending on interviewees to acquire their best responses. The interviewees were communicated via social media, informed about the purpose of the study, and asked if they would be willing to participate. Three SHP owners, a county government administrator, a provincial government administrator, as well as a hydro-ecology engineer agreed to join. The three SHP owners and the administrator were involved in pilot work for this project in 2014. One owner had previously retrofitted his facility with sluices and installed an ecological generator, and the other two owners had their projects decommissioned. The administrator had long-term experience in SHP management. The hydro-ecology engineer was selected as a representative member of the public. Interviews were recorded when permitted. The semi-interviews lasted 20–40 min and covered details of the SHPs of the interviewed owners as well as attitudes and perspectives on the challenges of E-flows implementation. The questions for SHP owners, the administrator and the engineer consisted of 6 close-ended questions and 3 open-ended questions; 3 close-ended questions and 3 open-ended questions; and 2 close-ended questions and 4 open-ended questions, respectively. Those questions covered attitudes and perspectives on the challenges of E-flows implementation and the issue of "who needs to pay for the losses" (see Tables S2–S4).

*2.3. Secondary Data Collection*

Secondary sources were used to determine environmental impacts as well as the losses in electricity production caused by E-flows implementation and the average returns of SHPs. Sources included the SHP Annual Statistical Report (2016) in Fujian Province and the 2017 Survey report on the status of rural hydropower projects in Fujian Province. Information and data were also obtained from the Fujian Provincial Department of Water Resources.

## 3. Results

*3.1. Environmental Impacts*

The results from the questionnaires showed that 20% of government administrators had once received letters of complaint related to dry watercourses caused by SHPs from local residents. This ecological impact was evidenced by government reports, with more than 93% (5815) of the projects resulting in dry reaches accounting for a total length of 7508.5 km, and around 7% (430) of the projects cut flows of up to at least 3 km of dry reaches (Figure 1).

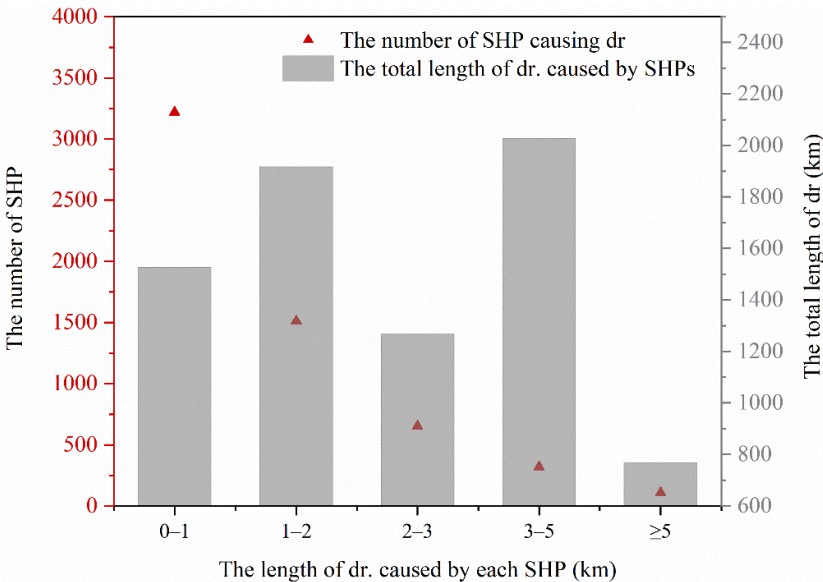

**Figure 1.** Lengths of dry reaches (dr.) generated by small-scale hydropower projects in Fujian Province. Source: [46].

In addition to dry river reaches, the excessive construction of SHPs in Fujian Province had turned some river reaches into reservoirs. Data collected from the Fujian Provincial Department of Water Resources show an average SHP spacing of 13 km on 65 rivers with a drainage area > 500 km², which has resulted in large decreases in discharge and flow velocity, which is conducive to algae growth. For example, on the Jiulong River, 10 SHPs operate on the trunk reach with an average spacing of < 7 km and the smallest spacing of just 5.4 km.

### 3.2. Conflicts

The initial attempt to implement E-flows regulation in Fujian Province encountered much resistance, with economic factors identified as the main obstacle, followed by stakeholders' skepticism and technical difficulties (Figure 2).

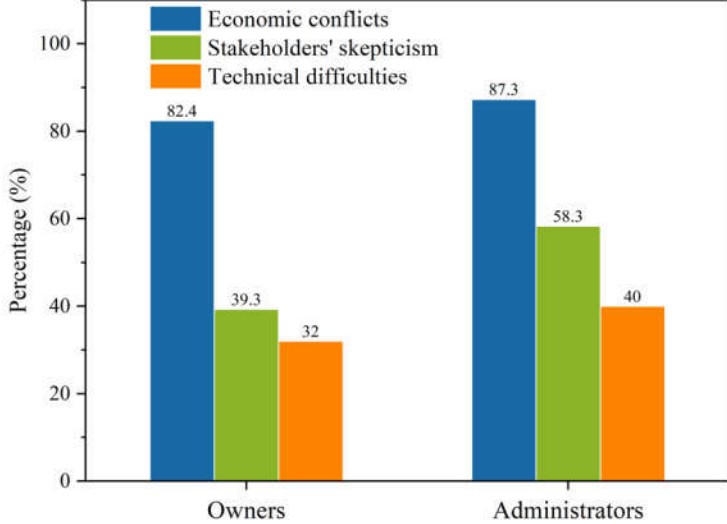

**Figure 2.** Opinions of small-scale hydropower project owners and government administrators on the difficulties of implementing E-flows (select one or more answer choices).

### 3.2.1. Stakeholders' Skepticism

All respondent groups tended to be positive about "whether SHPs are green energy", with support from 96.4% of the owners, 90.9% of the administrators, and 81.8% of the public. However, of the corresponding groups, 51.7%, 69.1%, and 81.9% considered SHPs to have negative impacts on the environment, respectively. Similar proportions (54.6%, 63.3%, and 91%, respectively) supported the implementation of E-flows.

### 3.2.2. Economic Conflicts

Because the administrators required the SHPs to release E-flows without any compensation, SHP owners were not willing to follow the requirement. Implementing E-flows involves reducing the discharge volumes available to produce electricity, which inevitably results in economic losses for SHP owners.

The SHP owner and SHP administrator groups were asked about the magnitude of losses experienced with the 10%MAF strategy, the results of which are shown in Figure 3. Nearly two-thirds of the owners of diversion-type SHP and nearly half of the owners of barrier- and mixed-type SHP suggested that their losses would exceed 10%. The diversion-type SHP owners estimated losses to be more than 15%, and these estimates were much higher than those of the barrier- and mixed-type SHP owners.

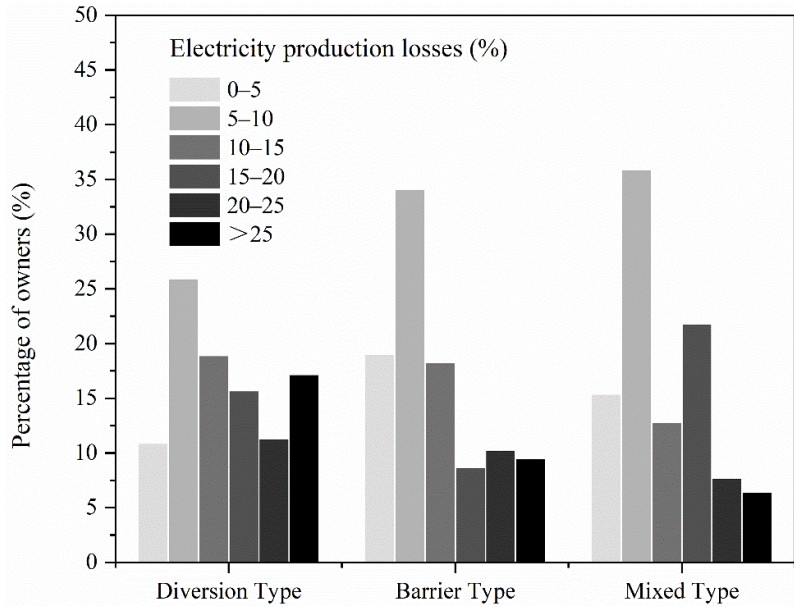

**Figure 3.** Influences of releasing E-flows on electricity production losses estimated by the owners of different types of small-scale hydropower projects.

These estimates are corroborated by our calculations of electricity production losses assuming the 10%MAF method based on available information, which implies that losses vary between SHPs types (Table 1). For example, impacts on diversion-type projects ranged between 9.7% and 23.6%. Overall, the calculated losses tended to decrease with SHP capacity; for barrier-type projects, production losses ranged from 3.6% to 8.6% and decreased in line with the single installed capacity of the SHPs. In general, reservoir-type projects only have generators with large capacities, while run-of-river projects usually have more generators with different installed capacities. Therefore, reservoir-type projects typically suffer comparatively higher production losses. The losses of mixed-type projects vary from 9.7% to 11.7% of their expected electricity production. Both official data (Table 1) and the questionnaire responses show that different types of the SHPs are subject to

varying production losses as a result of E-flows regulation, with diversion-type (accounting for 76.7% of the projects in Fujian Province) being most affected.

**Table 1.** Influence of releasing E-flows on electricity production losses of small-scale hydropower projects. Source: [47].

| The Name of SHPs | SHP Type | Catchment Area | Installed Capacity | Reservoir Storage | 10%MAF | Electricity Production Losses |
|---|---|---|---|---|---|---|
| | | km² | MW | 10⁴ m³ | m³/s | % |
| United Huiji | Diversion type | 63 | 0.50 | 247.0 | 0.182 | 12.6 |
| Lutouxia | Diversion type | 285 | 2.50 | 49.1 | 0.820 | 16.4 |
| Dongxiwei | Diversion type | 5 | 0.25 | 3.0 | 0.0013 | 23.6 |
| Yongxi | Diversion type | 224 | 40.00 | 6900.0 | 0.795 | 9.7 |
| Longmeishan | Diversion type | 22 | 1.00 | 14.5 | 0.060 | 16.8 |
| Sixth Cascade Project of Qingyin River | Diversion type | 329 | 7.500 | 4400.0 | 0.930 | 10.9 |
| Shanzai | Barrier type (Reservoir type) | 1646 | 33.00 | 17,600.0 | 5.700 | 8.6 |
| Shangjishan | Barrier type (Run-of-river) | 1138 | 2.80 | 61.0 | 3.570 | 3.6 |
| Dongxi | Mixed type | 42 | 3.20 | 189.9 | 0.215 | 11.7 |
| Yangmeizhou | Mixed type | 128 | 11.30 | 201.0 | 0.253 | 11.5 |
| Fuquanxi I | Mixed type | 116 | 8.50 | 1758.0 | 0.380 | 9.7 |
| Fuquanxi II | Mixed type | 158 | 36.10 | 337.0 | 0.730 | 10.16 |

An analysis of the questionnaire responses from SHP owners and administrators relating to estimates of electricity production losses and average returns is presented in Figure 4. The SHP owners estimated slightly higher losses than the administrators, and more than one-third and one-fifth of the owners and administrators believed that E-flows accounted for 15% of losses, respectively. One-third of the administrators believed the losses were low (0–5%), while only one-seventh of the owners believed this was the case (Figure 4a). Overall, the administrators were more optimistic than the owners, with approximately one-third believing that SHP received > 10% profit compared to one-fifth of the owners (Figure 4b). Despite differences in opinions on average returns of SHPs, there was no marked difference in estimates of electricity production losses ($p > 0.05$), with both groups suggesting relatively losses overall (Figure 4a).

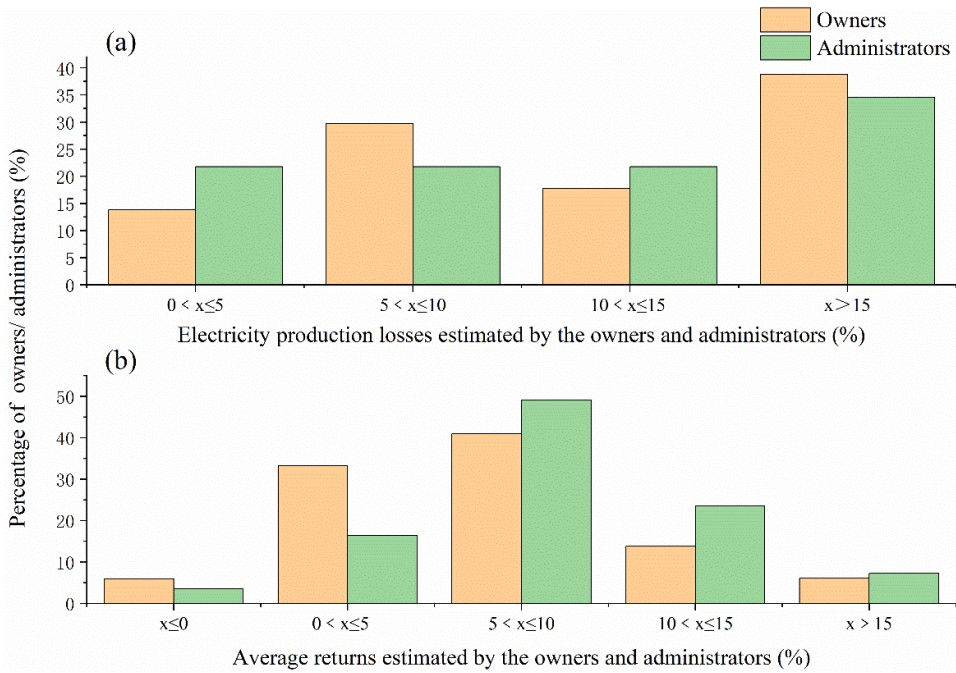

**Figure 4.** Estimates of (**a**) electricity production losses overall and (**b**) average returns of small-scale hydropower projects by owners and administrators.

It is known that profits of SHPs will be lowered to several levels after releasing E-flows, particularly for diversion-type SHPs. Because the SHPs have E-flows releasing infrastructure, most SHP owners have to pay for retrofit costs. Without subsidy, the costs of releasing E-flows could only be covered by the owners, and this would worsen conflicts, resulting in their unwillingness to follow the regulations.

### 3.2.3. Other Difficulties

In addition to economic issues and perspective divergences, E-flows policies are impeded by technical difficulties and management weaknesses. Although technical issues ranked last amongst the potential challenges (Figure 2), the installation of online monitoring facilities remains technically difficult. For example, wireless signals are poor in remote areas, making it impossible to perform online network monitoring. In the case of governmental management, 66% of the public did not believe that owners would release E-flows without strict government supervision even if they were given proper compensation.

*3.3. Approaches to Mitigating Conflicts*

3.3.1. Pilot Approaches

Due to the failure of the initial E-flows policy, the Fujian Provincial Department of Water Resources selected Changting and Yongchun counties as pilot sites to explore PES approaches to mitigating conflicts. In this program, SHP owners obtained extra on-grid tariff subsidies or one-off compensation payments based on the following approaches according to interview responses for illustration.

i. Retrofit works

Existing facilities were retrofitted to release E-flows, such as adding flow release holes and modifying sluices. The installation of ecological generators was also encouraged, which can utilize E-flows to generate electricity to reduce economic losses.

Xiyuan projects included a reservoir- and a diversion-type project, the latter being 1.5 km downstream of the former along a 2-km diversion channel and a 1.5-km dry reach. The Xiyuan reservoir-type project was later retrofitted by adding sluices in the diversion channel to release E-flows, and an ecological generator with a capacity of 0.125 MW was installed in the power plant of the reservoir project. The estimated losses caused by releasing E-flows (340,000 kWh) equate to approximately 0.095 million CNY, equivalent to an extra 0.07 CNY/kWh (on-grid tariff) to cover the losses. The additional cost for the ecological generator, sluice retrofit, and monitoring facilities was 0.46 million CNY. The government offers an extra on-grid tariff of 0.05 CNY/kWh as compensation as well as a subsidy of about 50% of the cost of the ecological generator.

ii. Restricted seasonal operation

These diversion-type projects, which cannot meet the E-flows needs, were prohibited from operating during the dry season (December–February). For example, the Qingyuan diversion-type SHP, which resulted in a 6.8-km-dry reach, was prohibited from running during the season. The estimated resulting production losses were 0.31 million kWh, equivalent to approximately 0.93 million CNY, with an additional on-grid tariff of 0.072 CNY/kWh needed to cover the loss. The cost of the sluice retrofit and monitoring facilities also exceeded 0.05 million CNY. Based on the PES scheme, the SHP owner received an extra on-grid tariff of 0.05 CNY/kWh as compensation.

iii. Decommissioning

SHPs that are too difficult to retrofit were decommissioned under the condition of guaranteed irrigation and public safety.

The downstream section of the Hongqi SHP area is a popular natural spot. However, due to the improper operation of the project, flows in the trunk stream were delivered to diversion channel, leading to a 2.4-km dry reach and significant damage to the landscape character. After prolonged negotiation, 2.52 million CNY (approximately 60% of the appraisal price of the project considering the installed capacity, electricity production, on-grid tariff, construction time, etc.) was paid as compensation for dismantling the project. The Hongqi project dam was eventually removed, although its power plant was retained as a hydropower museum.

Such practices have been successful in the study region by addressing the occurrence of dry reaches caused by SHPs. It is noted that instead of being dismantled, some power-plants have been converted into museums, cafés, or libraries, thereby providing beneficial public spaces for the neighboring communities. "The government's regulations" and "the obligation and responsibilities for the environment" were all mentioned by the interviewed SHP owners when asked why they finally agreed to implement E-flows or decommission their projects. Those owners who supported the implementation of E-flows emphasized that to avoid opposition, the government ought to fully consider stakeholders' interest. In addition to compensation and subsidies, government administrators also at-

tributed the success of these schemes to "constant communication and negotiation between the government administrators and the owners" as well as "the long-term publicity towards the importance of ecosystems".

### 3.3.2. Current Economic Incentives

In light of the success of the pilot PES scheme, the Fujian Provincial government required all SHPs to be installed with monitoring facilities by the end of 2020 [48]. To facilitate this, SHP owners receive various levels of compensation depending on the nature of the work, i.e., retrofit work, seasonally restricted operation, or decommissioning. The projects requiring retrofitting work are awarded an extra on-grid tariff of 0.02 CNY/kWh; those adopting seasonally restricted operation are subsidized by an extra 0.03 CNY/kWh; and for decommissioned projects, owners can receive 50% of the market price as compensation. By the end of 2019, 1966 projects had already implemented E-flows, and 584 projects had been decommissioned [49]. Because the energy supply in Fujian Province is sufficient, the losses in electricity production resulting from these schemes do not currently have any negative consequences for industrial production or the standard of living.

To date, retrofit works have been widely applied, although the owners of diversion-type projects have suffered relatively higher losses than those of barrier-type projects. Therefore, the fairness of the different PES schemes may become an issue. Based on the questionnaires, SHP owners adopting seasonally restricted operations did not consider the PES subsidy sufficient, with only 39.5% of the owners and 21.8% of the administrators supporting this scheme. Indeed, only approximately 10% of the SHP owners and administrators were satisfied with current economic incentives, while approximately half of these two groups expected incentives to be scaled based on relative economic losses.

## 4. Discussion

### 4.1. Improving PES Programs

Individuals adversely affected by environmental policies need to be sufficiently compensated [50]. In the case of E-flows, relatively low levels of compensation will influence the sustainability of policies, as although owners may reluctantly release E-flows in the short run under pressure from the government, less effort will be given to maintenance and management of E-flows over the longer term.

Furthermore, the calculation of E-flow impacts varies depending on which methods of assessment is adopted. Furthermore, losses in electricity production vary between SHPs —even under the same hydrological circumstances—depending on which approaches are taken [51]. However, current compensation strategies do not reflect the actual losses incurred by owners. Thus, it is more reasonable to apply differentiated compensation based on SHP electricity generation losses rather than E-flows. Therefore, a price system based on differential compensation according to the actual electricity production losses incurred to maintain E-flows is recommended.

Apart from the amount of compensation [52], the source of compensation is a prickly issue to tackle. A long-term, funding-supported system should be established as soon as possible [29]. The benefits of restoring river ecosystems are well known and all beneficiaries must bear some responsibility. As the direct parties involved, SHP owners should take the initiative to undertake environmental improvements, whereas consumers, as indirect parties, need to take responsibility for triggering the demand for environmental services. Therefore, both parties should bear some of the losses caused by implementing E-flows. The government can offer subsidies for retrofitting dams, installing ecological generators, and monitoring facilities. Considering that government finances may not be able to afford ongoing compensation, we propose a cost-sharing PES program paid by all interest groups. Similar to thermal power, for which on-grid tariff includes the costs of denitration, there is an opportunity to recover the partial costs of releasing E-flows from some electricity consumers. The raised on-grid tariff of hydropower was still the lowest among all

types of energy in Fujian Province, at 0.33 CNY/kWh compared to 0.39 CNY/kWh for thermal power [53], 0.4 CNY/kWh for nuclear power [54], and 0.48 CNY/kWh for wind power [55]. Based on our questionnaires, the option of sharing the additional costs of implementing E-flows between government, owners, and electricity consumers gained the highest level of support among each group (Figure 5), which suggests the potential for establishing such a PES program.

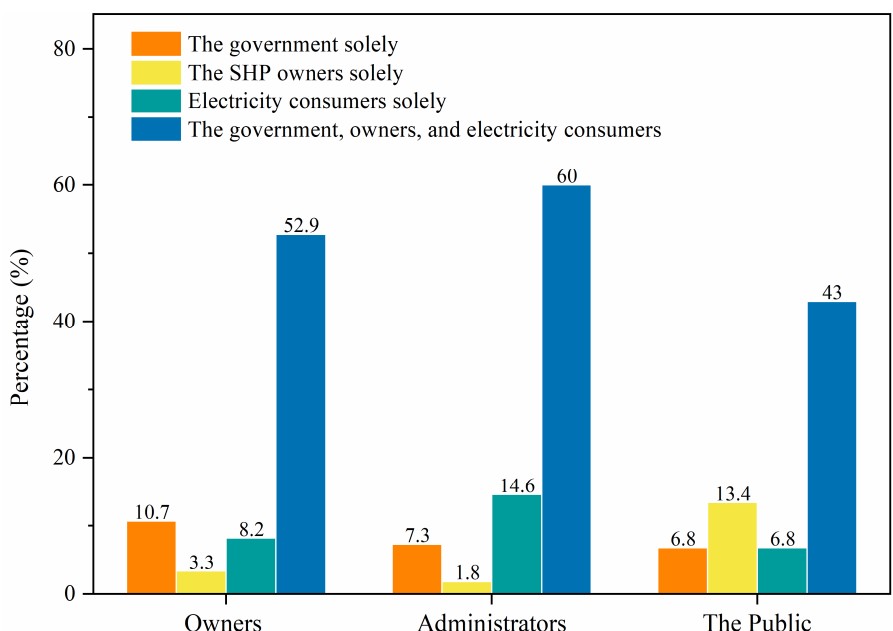

**Figure 5.** Comparison of the views of short-term hydropower project owners, administrators, and the public on a cost-sharing PES program for E-flows (select one or more answer choices).

Based on the electricity generation by different types of SHPs, if all SHPs adopt the 10%MAF strategy when implementing E-flows, the average electricity production losses are estimated to be approximately 9.38% of the total electricity production of SHPs in Fujian Province. Considering the average annual electricity generation (approximately 23.3 billion kWh [56]) and on-grid tariff of SHP in Fujian Province, the losses of releasing E-flows are calculated as 2.18 billion kWh, which amounts to approximately 721.22 million CNY. According to the total electricity consumption (211.27 billion kWh [57]) and the average on-grid tariff (0.6 CNY/kWh) in Fujian Province, the total electricity consumption costs 126.762 billion CNY. The impact ratio of electricity bills is derived from losses of releasing E-flows divided by total electricity consumption costs, which is calculated as 0.57%.

If all the losses caused by implementing E-flows are transferred to the electricity rate, the impact on people's original electricity bills is approximately 0.56%. On the basis of the cost-sharing principle, electricity consumers and the SHP owners would bear this impact together; if this system was adopted, electricity consumers pay < 0.56% more than their existing electricity bills. The results from the questionnaires show that although there is significantly less support from administrators than the public, nearly three-quarters of both groups were willing to pay 1% more than their usual electricity bills to support E-flow implementation.

### 4.2. Improvement of Communication and Management

The cooperation of stakeholders is essential for successful E-flows implementation. In general, the greater the acceptance of the need for E-flows, the more likely a successful partnership can be formed. Our results indicate that the perception of the necessity of E-

flows differs between and within interest groups. Thus, all groups need to increase their eco-awareness of the need to achieve environmental protection. Ensuring appropriate that communication during the decision-making process will further ensure the success of implementing scheme [58]. This is crucial to enable all stakeholders to raise and resolve potential disagreements [59]. Our study showed that communication is an essential component of collaboration; active dialog between interest groups helps to reach a compromise, allowing potential conflicts to the recognized and addressed during the implementation process.

Moreover, understanding of SHPs is often one-sided, largely depending on where their benefits lie. SHPs are generally welcomed, as they are the cheapest and most accessible means of obtaining electricity [60,61]; however, with improving living standards, Chinese residents have begun to pay more attention to environmental quality. Indeed, E-flows schemes have little negative impact on the public's economic interest but bring environmental and recreational benefits, which may account for their relatively low level of recognition of "SHPs belong to green energy" and high level of support for E-flows implementation.

Monitoring the long-term impacts of current measures is also helpful for informing subsequent management [44]. As there remain unknown relationships between flows and biotic responses [62], monitoring is needed to address this uncertainty [63], and local electricity users can be successfully involved in this monitoring work [64]. Additionally, publishing the outcomes of current monitoring measures should help bolster public support [65], which would likely enhance public desire for further E-flows implementation. Specific E-flow assessments could be conducted on SHPs located in ecologically sensitive regions in light of the capacity and available resources of regional and local governments. Undoubtedly, gradually augmenting the scale of E-flows implementation seems inevitable, which must be matched by suitable compensation schemes.

As people's environmental requirements have changed, government understanding and regulation of water resources need to change too [16]. Future water resources planning should strive for both comprehensive and coordinated development of the environment and society. Taking environmental factors into account at the planning stage will help identify potential stakeholder conflicts that will otherwise need to be tackled at a later date.

## 5. Conclusions

E-flows have been recognized as a crucial water management tool when aiming to meet both environmental and societal needs. This study represents, to the best of our knowledge, the first attempts at exploring solutions to mitigate the conflicts in E-flows implementation for SHPs based on questionnaires and interviews of three interest groups. We used Fujian Province as a case study to demonstrate the challenges facing E-flow implementation, focusing on (1) skepticism about "whether SHPs are green" and "the necessity of releasing E-flows" among SHP owners, government administrators, and the general public; (2) economic conflicts caused by electricity production losses especially in the case of diversion-type projects; (3) inadequate governance; and (4) PES. Importantly, our questionnaires and interviews reveal that there is potential for establishing a long-term cost-sharing PES program, paid by the government, SHP owners, and electricity consumers and emphasize that successful E-flows implementation will benefit from sustained and effective communication between all interest groups.

As E-flows enter the implementation phase, it should be recognized that economic challenges remain the strongest driver and key obstacle to implementing environmental policies [66]. Furthermore, it is worth recognizing that while E-flows implementation is a valuable tool, this is not the only measure available for river rehabilitation concerned with SHPs. For example, fish pass facilities need to be established to improve longitudinal continuity. While beyond the scope of this study, further work is also needed to consider the ecological responses to E-flows schemes so that they can be enhanced and optimized in

the future. This requires the cooperation of scientists and water managers [67]. Finally, we emphasize that a combination of social, economic, and environmental disciplines is needed to enhance existing understanding and overcome the potential challenges of implementing and managing E-flows schemes.

**Supplementary Materials:** The following are available online at www.mdpi.com/2073-4441/13/18/2461/s1, Table S1: Questionnaires sent to SHP owners, SHP government administrators, and the public; Table S2: Interview protocol questions for SHP owners; Table S3: Interview protocol questions for SHP government administrator; Table S4: Interview protocol questions for hydro-ecology engineer.

**Author Contributions:** Conceptualization, Q.R.; methodology, Q.R.; validation, F.W.; investigation, Q.R.; resources, Q.R.; writing—original draft preparation, Q.R.; writing—review and editing, W.C.; supervision, W.C.; project administration, W.C. and F.W. All authors have read and agreed to the published version of the manuscript.

**Funding:** Funding: This research was funded by the National Natural Science Foundation of China (41771500) and the Environmental Protection Technology Program of Fujian Province (2021R023).

**Institutional Review Board Statement:** Not applicable.

**Informed Consent Statement:** Informed consent was obtained from all subjects involved in the study.

**Data Availability Statement:** Data are contained within the article and Supplementary Material.

**Acknowledgments:** Our sincere thanks go to survey respondents for patient answers, to Nina Morris from the University of Edinburgh, and Baoli Liu from Fuzhou University for their valuable advice.

**Conflicts of Interest:** The authors declare no conflict of interest.

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
