# Peer review of "Conflicts in Implementing Environmental Flows for Small-Scale Hydropower Projects and Their Potential Solutions—A Case from Fujian Province, China"

_water, doi:10.3390/w13182461_

Round 1
Reviewer 1 Report
The authors present an interesting case study of SHPs in Fujian Province and the results of a perception-based survey administered to SHP operators, government administrators, and the general public. The main topic, environmental flow maintenance and enforcement, is timely and important, and the authors uncovered some interesting information through the survey. But there are a number of issues that I believe must be addressed before this manuscript could be considered further.
Although the authors state clearly that the focus is on environmental flows, they seem to be glossing over the other ecological impacts. The study they cite (Bakken et al. 2012) does state that 100% of the 27 SHP projects they assessed reported a reduction in flows, but this hardly constitutes a settled matter for SHPs worldwide. Numerous studies have come out since, from many countries (including China) that highlight issues such as the loss of connectivity and habitat, not to mention the network effects of SHP cascades. This does not invalidate the authors’ findings, as environmental flows are rightfully an important concern, but I believe some additional context and updated references are needed in the Introduction, and a more thoughtful discussion of the limitations of addressing e-flows in isolation.
It is also necessary to include a short background on the hydroelectricity market in China. What is required to get a SHP licensed, permitted, and operational? Must all electricity generated be purchased by the grid (there was a mention of Fujian province not needing all of the generation)? Presumably the government is regulating electricity rates, and so what is the scope and process for raising rates? I admit I was struck by the idea that many of these SHPs are probably unnecessary and decommissioning might be the best solution, but it is difficult to know without some more context.
Materials and Methods section is too short. More needs to be said about the surveys and interviews, but there is nothing mentioned about the environmental impacts analysis, which appears to have been done using actual hydrologic data (e.g., Figure 1).
-45-46: this sentence is unclear, but the reference is also almost 20 years old and needs updating; much has been written since 2002 on the role of e-flows as well as on SHPs and their impacts
-65: “more frequently” compared to what? Other provinces? Or historical patterns?
-74-75: can this be expressed as a percentage of the total SHPs that should have facilities installed? Earlier in the paragraph there is mention that twelve primary rivers require it, so presumably it is a subset of the >6000 SHPs in the province, but it would be good to know how large the implementation gap is between the 118 SHPs that complied with the monitoring requirement and what is currently mandated
-76-79: this paragraph should be expanded, to introduce the methods that will be used, the reasons for the authors’ hypothesis, and what they mean by cognitive and technical factors
Section 2.1: please say more about how survey respondents were selected. For example, were members of the public targeted based on their proximity to a SHP? The response rate appears extremely high across all respondent groups—is there an explanation for this? And it would also be helpful to include a sample question or two—I see that the Questionnaire is available as Supplementary Material, but I do not have access to this, and it is still important to help the reader understand, for example, if questions used rating scales, categorical responses like multiple choice, etc. without making them consult the Supplement.
Section 2.2: say more about why the interviews were conducted, and how interviewees were selected. Was it at the end of the survey? Presumably they were administered to capture qualitative data. But why so few interviews?
Figure 1: where do these data come from?
Figure 2: this is impossible to interpret without understanding the specific questions asked. And again, ‘cognizance conflicts’ and ‘technical difficulties’ need to be defined somewhere.
Section 3.2.1: this section is very difficult to understand. The authors have still not defined ‘cognizance conflicts’, and it is unclear 1) how questions were asked, 2) how respondents were able to answer (gradient scale, binary response, etc.), and 3) how some of the questions relate back to E-flows
Figure 3 needs a different color or gradient scheme, or perhaps another way to present the data that is easier to read and interpret
Table 1 and related paragraph: methods were never described!
Figure 4: this is a confusing way to present the data, because having both average returns and electricity production losses on the same bar suggests they are correlated when of course, the inverse is more likely. There is no reason to report them on the same graph in this manner.
-177-179: in what way do your data justify this statement, that there is “no marked difference between their calculations of electricity production losses”?
-196-198: I don’t understand this statement, about divergent attitudes among administrators creating “resistance to supervision”
-198-199: is this based on the survey? Was there a question about overlapping jurisdictions? If so, please clarify, otherwise this statement needs some other kind of substantiation. It is too vague as written.
-Section 3.3 does not fit well in the Results section. I understand that some of the data reported are derived from interviews, but there is also a lot of case study background included, as well as more detail than is necessary on the individual projects. And Section 3.3.3 is more appropriate for Discussion (Section 4)
-Section 4.2’s first paragraph is vague and it is difficult to discern any clear takeaway or discussion point. It includes a lot of general statements that may or may not actually support e-flow implementation, and are not clearly linked to the study.
-362: why do local electricity users need to be involved in monitoring?
-375: this statement about e-flow implementation alleviating impacts is too strong; it may alleviate some impacts.
Author Response
Dear Reviewer,
Please see the attachment of our responses.
Best Wishes,
Qizhen Ruan, Feifei Wang, Wenzhi Cao

Reviewer 2 Report
General Comments:
- The authors need to do a thorough proof reading of the entire manuscript, since there are some areas that need grammar correction and spell checks.
- Put more emphasis on the contribution section on the second page. What really is the manuscript trying to recommend and contribute to the existing debate around this issue? Are the responses similar to studies done or if this is the first pilot study done- the authors need to mention that clearly and summarize their contribution.
- The materials and methods section is brief and does not present the results of any test conducted or elaborate the process of interview and selection of the cohort of interviewees. At least a table could be shown to illustrate the region, the selection and the number of observations from each group.
- The major problem that I found was the absence of an elaborate description of the PES scheme referred to in the paper. The authors highlight it as a recommendation going forward on implementation of E flows. Is there a procedure used for estimating the increase in electricity bills for consumers? The details should be worked out.
- Some of the graphs are not clear (Fig 5 for instance).
Author Response

(The authors gave the same response as above.)

Reviewer 3 Report
I thank you for this interesting study with cogent arguments why economic policy instruments like PES need to be applied to systematically make environmental flows, part of doing business on rivers. I particularly like the ideas presented that seek to fund the payments through electricity user charges - which internalises the cost. overall i am impressed with the manuscript and see it as a study worthy of publication. There are however, numerous minor improvement required which are to do with syntax and English language writing. A good proof reader and copy editor should be able to assist in ironing out these minor problems
Author Response
Dear Reviewer,
We are appreciated that you are interested in our topic and the positive comment for our manuscript. E-flows implementation for SHPs is at the beginning stage in China and much work needs to be done in the future. We are very sorry for the mistakes in this manuscript and the inconvenience they caused in your reading. The manuscript has been thoroughly revised and edited by a native speaker, so we hope it can meet the journal’s standards. We have also carefully made some changes according to other reviewers' suggestions. Once again, we thank you for the time you put into reviewing our paper and look forward to meeting your expectations.
Best Wishes,
Qizhen Ruan, Feifei Wang, Wenzhi Cao
Round 2
Reviewer 1 Report
I commend the authors on a thorough revision and careful attention to previous comments. My remaining comments are minor and expect that they will not be difficult to address:
-12: suggest “To understand those obstacles” rather than “overcome”
-16: I understand what the authors mean by “divergent opinions” but am not sure it will be clear enough to readers who will first see it in the abstract, with no explanation. I suggest the authors consider an alternate term, like “skepticism about effectiveness” and use that throughout the text.
-19: I think “10% mean annual flow” needs to be explained—I interpret it as an upper limit for flow deviation, but as written it could be misunderstood as a minimum target for the stream.
-52-53: This is not a complete sentence
-54: the last 50 years? Unless you are including a reference from 1970, I don’t think you can make such a claim. I would suggest that the last 20 years is more reasonable, as that is the period corresponding to a real increase in studies and legislation.
-60: not clear what “mainstream E-flow literature” means
-81-82: 10% MAF and Qdm90 both need to be described, as they might be unique to China and not immediately understood by other readers or non-hydrologists
-92: “stakeholder perspectives” is vague, and per my earlier comment, it would be more clear to use a term such as “skepticism” to note that there is some kind of mistrust or disbelief about the impacts of SHPs.
-section 2.1—the revised section is much more clear about how the survey was administered. Were any demographic data collected? If so, it is common practice to provide summary data, to help readers determine how representative the sample was. You should also say just a bit more about how initial recipients were targeted, and whether they had the ability to share the link with others (i.e., a ‘snowball’ sampling approach). In any case, it sounds like a non-probability sampling approach, which is okay but should be noted because it influences how one interprets the data.
-Table 1—it is good to include the sample questions but I recommend keeping them within the text, because it is hard to read them in the Table format.
-Figure 2. I suggest including the question that was asked here, if possible. When percentages do not total 100, it is clear that there is some kind of multiple choice, but then is there any sort of ranking of responses? Or is it just that most people said “Yes” to economic conflicts, and then smaller proportions said “Yes” to the other two factors?
-223: word missing here—“relatively ? losses overall”
-238: I think this should state “…did not believe that owners release E-flows…”
-326: is ‘ecological energy’ a translation of a term already in use in Fujian province? If not, it is confusing and I would recommend not proposing a new term here. It suffices to say “a price system based production losses incurred to maintain E-flows” or something like that.
-355-358: The math is difficult to follow here. I gather that 721.22 million CNY is based on the 0.33 CNY/kWh tariff noted on line 341. So why is the tariff listed at 0.6 CNY/kWh on line 357, and how was the final calculation 432.73 million CNY??? And what figures are being compared to determine the impact would be <1% of electricity rates?
-374-377: I don’t think you can claim your study showed this; at least, it does not follow from your methods or the data and results you presented
Author Response
Dear reviewer,
Please see the attachment.
Best Regards,
Quizhen Ruan, Feifei Wang, Wenzhi Cao

Reviewer 2 Report
The current version is vastly improved from the previous version . Proof reading is still needed and could still elaborate on the contribution and methodology but not significant changes are required.
Author Response
Dear reviewer,
We really appreciate your positive comment on our last version of the manuscript. The valuable suggestions you proposed helped us to raise the quality of the manuscript. We have carefully considered your and other reviewers' suggestions and revised the manuscript accordingly. In what follows, we would like to answer the questions you mentioned and give a detailed account of the changes made to the original manuscript.
Contributions of this work have been restated in the abstract as well as the beginning of the conclusion. The revision could be found in lines 12, and 427-429.
We also elaborated on the sampling approaches of questionnaires as well as the questions of interviews and the limitations of the method we used. The revision could be found in lines 108-165.
We are very sorry for the grammatical mistakes in this manuscript. The manuscript has been thoroughly proofread again and we hope it can meet the journal’s standard this time. Once again, we thank you very much for the time you spent reviewing our paper and look forward to meeting your expectations.
Best Regards,
Qizhen Ruan, Feifei Wang, Wenzhi, Cao